# Torque Teno Virus (TTV)—A Potential Marker of Immunocompetence in Solid Organ Recipients

**DOI:** 10.3390/v16010017

**Published:** 2023-12-21

**Authors:** Agnieszka Kuczaj, Piotr Przybyłowski, Tomasz Hrapkowicz

**Affiliations:** 1Department of Cardiac, Vascular and Endovascular Surgery and Transplantology, Faculty of Medical Sciences in Zabrze, Medical University of Silesia, 40-055 Katowice, Poland; piotr.przybylowski67@gmail.com (P.P.); thrapkowicz@sum.edu.pl (T.H.); 2Silesian Center for Heart Diseases, 41-800 Zabrze, Poland

**Keywords:** Torque Teno Virus, organ transplant, immunosuppression

## Abstract

Torque Teno Virus (TTV), first discovered in 1997, is a non-pathogenic, highly prevalent virus with a notable presence in the human virome. TTV has garnered attention as a potential indicator of immunocompetence in recipients of solid organ transplants. In this review, we discuss the role of TTV as a potential marker for immunosuppression optimization, prediction of graft rejection, and as an indicator of opportunistic infections. We discuss TTV’s behavior over the course of time after transplantation, TTV’s implications in different immunosuppressive regimens, and potential utility in vaccinations. The review synthetizes findings from various studies depicting its potential clinical utility for future personalized patient care.

## 1. Introduction

TTV was first discovered in Japan in 1997 in patients with hepatitis after the transfusion of blood products. Currently, it is assumed that the virus does not cause any significant disease. The potential virulence of the virus is not known, including in immunocompromised hosts. It belongs to the family Annelloviride, genus Circovirus [1]. It is a non-enveloped, negatively polarized, single-stranded circular DNA virus comprising 3.8 Kb nucleic acids [2]. The virus occurs in other primates and animals [3]. The prevalence of TTV increases with age and reaches a plateau in early childhood. It causes chronic infection with no established disease or clinical manifestation. The incidence among blood donors reaches up to 100% [4]. The virus is transmitted by many means, including parenterally and enterically [5]. It occurs in serum, in white blood cells (lymphocytes, granulocytes), saliva, and on mucosal surfaces. It can be detected in liver tissue and bile duct epithelium [6], gingival tissues [7], vascular walls [8], and many other tissues. TTV comprises the majority of the human virome, both in terms of percentage and the number of copies [2]. The current methods of assessment in humans comprise PCR detection methods in serum samples. The PCR methods rely on the replication of highly conservative DNA sequences in the so-called untranslated region [2]. TTV replication tends to be higher in immunocompromised hosts. In the groups of organ recipients, over 85% were positive prior to liver or renal transplantation [9,10]. Before lung transplantation, TTV was detectable in of 93% patients [11]. Among heart transplant recipients, the available studies, albeit limited, indicate TTV presence in nearly all patients [12]. In children after renal transplant (NTx), the prevalence was as high as 94.5% [13]. Observations have shown that the number of copies increases after transplantation [9,10,14,15,16,17,18]. What is especially promising about the role of TTV as a marker of immunosuppression and the risk of graft rejection is that the prevalence and number of copies are unaffected by the current antiviral therapy and prophylaxis used in organ recipients [2].

Currently, there are many evolving methods assessing the probability of graft injury [19]. Donor-specific antibodies are connected with antibody-mediated rejection, future graft failure, risk of vasculopathy, and future risk of death and retransplantation among recipients [20]. Studies have shown that gene expression profiling with commercially available and accepted methods may exclude cellular rejection, with high probabilities in groups with low pre-test probability of rejection [21,22]. In turn, donor-derived cell-free DNA informs about the current graft injury caused by many factors (acute cellular rejection, antibody-mediated rejection, graft ischemia, and others not directly associated with the graft) [23]. However, we do not have accessible techniques to assess the strength of applied immunosuppression besides immunosuppressive drug level monitoring. We still rely on clinical course (assessment of infections, surveillance of graft function, monitoring of adverse drug effects) and results of biopsies.

Immunosuppressive drugs are necessary for graft survival, but their application is connected with opportunistic infections, cancers, metabolic disorders, and cardiovascular complications. The drugs have a small therapeutic window, and their appropriate drug level monitoring corresponds more with avoiding drug adverse effects than with an appropriate grade of immunosuppression. The achievement of the therapeutic drug level is not always connected with freedom from rejection [24].

## 2. Materials and Methods

We conducted a comprehensive search across several databases, including PubMed, ScienceDirect, Scopus, and Google Scholar, using keywords such as Torque Teno Virus (TTV), transplantation, solid organ transplantation, immunosuppression, infection, and vaccination. After excluding duplicates, we found 180 studies. Enrolled were studies with TTV detection relying on PCR tests: in-house real-time PCR tests and commercially available tests. In further analysis, we excluded hematopoietic stem cell transplantation and tumor patients without transplantation. We identified 86 studies (excluding state-of-the-art analyses and meta-analyses) focusing on patients after solid organ transplantation in the context of Torque Teno Virus. We included articles relevant to our research area, without a specific timeframe. Selection criteria were based on the significance to the content, relevance of the topic, involvement of transplanted organs, clinical importance, and relevance to human subjects.

## 3. Available Methods of Immunosuppression Optimization—The Potential Role of TTV in Immunosuppression Management

In the case of heart transplant recipients, invasive monitoring of graft rejection with endomyocardial biopsies is still mandatory to avoid graft loss in high-rejection-risk recipients [20]. On the other hand, too high immunosuppression, not seen in endomyocardial biopsies, is connected not only with infection, but also impaired control of carcinogenesis. The generally accepted rules dividing patients into recipients with a high risk of rejection, like patients with a high level of panel of reactive antibodies, multiparous, patients after mechanical circulatory support, and low-risk patients like elderly individuals and patients with prior cancer, are too general [20]. Up to now, available analyses comprising function and subsets of lymphocytes have no relevant application in everyday practice [10]. The small non-pathogenic and highly prevalent viruses like TTV seem to be promising as a novel tool in recipient care. Figure 1 schematically illustrates the proposed utilization of TTV assessment in solid organ recipients. While the concept appears clear and promising, there are concerns and discrepancies regarding the actual implementation of TTV assessment in this population. The interest in TTV reflects the TTV Guide Tx study. The project, founded by the European Union, aims to achieve a 20% reduction in infections and graft rejection among NTx recipients while simultaneously enhancing graft and patient survival. Proponents of the project assert that these anticipated advancements will lead to a reduction in healthcare costs by approximately EUR 50 million. The initiative encompasses 20 European universities and institutions, commencing in May 2021 and spanning a 5-year duration, with the medical University in Vienna serving as the coordinating institution. The total cost of the project amounts to EUR 6,099,831.00 [25].

Also underway is a multicenter (currently comprising three participating centers), open-label, randomized interventional trial in lung transplant recipients. The VIGILung study (Viral Load-Guided Immunosuppression After Lung Transplantation) aims to optimize tacrolimus administration based on TTV load. Patients are randomly assigned to either the conventional tacrolimus dosing arm or the TTV-tailored dosage optimization arm. The study commenced in August 2020 and is expected to continue until 2025, with an estimated enrollment of 144 patients [26,27].

## 4. TTV Load—Correlation with Opportunistic Infection in Graft Recipients

TTV seems to be a marker of opportunistic infections in graft recipients. High viremia was a predictor of CMV reactivation in the graft recipients. Patients who developed CMV reactivation showed higher levels of TTV viremia regardless of the organ transplanted. In a mixed population of 280 renal transplant (NTx) and liver transplant (LTx) recipients, TTV viremia higher than 3.45 log/mL within 10 days after transplantation was higher in CMV DNA-positive compared to CMV DNA-negative individuals, and it correlated with subsequent CMV reactivation [10]. TTV load may be useful in predicting CMV reactivation and viremia control in seropositive renal graft recipients. Among 64 CMV seropositive NTx recipients, evaluating TTV load alongside CMV Quantiferon assay before transplantation and at 1-month post transplantation allowed for the stratification of CMV reactivation risk within the initial year post transplant. The cited study established a cut-off value for TTV at 1 month of 4.23 log_10_/mL [18]. Nevertheless, certain studies have indicated a lack of correlation between viral infections such as CMV and BKV infection with TTV viremia in recipients of renal grafts. In a comprehensive retrospective analysis involving 389 living donor NTx recipients, examination of blood samples collected within the initial year post transplantation revealed no significant correlation between TTV load and the presence of these two viruses [9]. However, most of the existing studies demonstrate an association between TTV and the incidence of opportunistic infections. In a study relying on 54 renal graft recipients, it was shown that TTV load was higher 1–3 and 3–6 months after transplant among renal graft recipients who suffered from infections [17]. In a study comprising 221 kidney transplant recipients, the viral loads at 1 month of observation were higher in patients who subsequently developed an infection. The authors stated the cut-off value predicting infection at a level above 3.15 log_10_ copies/mL (hazard ratio: 2.88 at 95% CI 1.13–7.36, *p* = 0.27) [28]. A viral load > 2.65 log_10_ copies/uL between 1 week and 1 month after transplantation predicted an increased risk of infection in NTx patients, with a sensitivity of 99.73% (*p* < 0.0001) and specificity of 83.67% [16]. In lung transplant recipients, it was demonstrated that TTV levels above 9.3 log_10_ copies/uL were associated with opportunistic infections [20,29]. A meta-analysis comprising pooled data from seven studies on NTx recipients showed a significant difference between viral load in patients who subsequently developed infection and patients with no infection, but based on specificity, sensitivity, and the area under the receiver operating characteristics curve, this was described as a poor discriminator [14].

## 5. Role in the Prediction of the Response to Vaccination

Immunosuppressed patients have a known weaker response to vaccination. Many studies show a weakened response to applied vaccines in organ recipients [30,31]. Available guidelines inform that it is better to vaccinate over a longer period of time after transplantation and rejection treatment, as the response to vaccination in the early period is low [32]. In this context, it seems that TTV may be a valuable marker in predicting the response to vaccination [33]. An analysis involving 103 lung transplant recipients, spanning from 4 to 237 months post transplantation and receiving mRNA vaccination against COVID-19, evaluated the humoral response to the vaccine. Among the patients investigated, spike-specific IgG antibodies, indicative of humoral response, were detected in 40% of individuals following the administration of the second dose of the mRNA vaccine. The findings indicate a potential correlation, demonstrating that a higher TTV titer might predict a diminished humoral response to SARS-CoV-2 vaccination [34]. Within a study involving 94 TTV-positive NTx recipients, the initial viral load demonstrated a correlation with spike-specific IgG antibody levels 28 days post vaccination. Upon further analysis, the authors indicated that this correlation was statistically significant solely among patients within a 24-month post-transplantation period. However, in individuals with a longer interval since NTx, this correlation was not observed [35].

## 6. Changes concerning the Torque Teno Virus in Transplant Patients over Time

Among multiple studies comprising viral kinetics after transplantation in multiple solid organ recipients, one repeating scheme is visible. During the first 2 weeks after transplantation, the kinetics is not correlated with pretransplant viral load, and in some patient cases, it even decreases compared to the pretransplant value [11]. After this period, the viral load gradually increases, reaching a plateau 3 months after transplantation, and remains constant up to 1 year [10,12,18].

In a study designed to evaluate TTV dynamics during the initial 3 months post transplantation, 46 lung transplant recipients were monitored. TTV was detectable in 93% of patients before transplantation. Subsequently, in all recipients, TTV levels increased post transplantation, reaching peak levels at a median of 67 days following the procedure. The median peak values observed were 9.4 log_10_ copies/mL [11].

A multivariable analysis involving 94 NTx recipients revealed an independent association between time and the duration of post-transplantation period, indicating that the highest viral values were observed in the early stages following transplantation. At a median of 41 months post transplantation, the median viral load recorded was 3.78 log_10_ copies/mL [11]. Among 50 heart transplant recipients monitored during routine outpatient visits, a notable increase in viral load was observed post transplantation. The peak viral load was attained at a median of 121 (92–185) days after transplant, with mean values rising from 4.4 log_10_ copies/mL before transplantation to 9.39 log_10_ copies/mL during follow-up observations [36]. Similarly, in the two-center population of heart transplant recipients (106 patients), the highest TTV loads were observed 3 months after transplantation and reached a mean value of 8.0 log_10_ copies/mL [12].

Following the first year, the number of TTV copies slightly decreases within 2 to 3 years after transplantation [34,37]. Within a study involving 715 NTx recipients, with a mean follow-up of 6.3 years post transplant, TTV was detected in 95% of patients. The highest TTV loads were observed between 6 and 12 months post transplant. Subsequently, a stepwise decrease in TTV load was noted after the second and third years [37]. Among 103 lung transplant recipients, with a mean post-transplant duration of 55 months, only 6 individuals tested negative for TTV. The highest TTV values (>6.5 log_10_/mL) were observed in patients with the shortest duration after transplantation (median of 17.5 months), while the lowest values (3.78 log10/mL) were recorded in the group with the longest post-transplant duration (median time of 97 months) [34]. The observation was similar for living donor renal graft recipients [9]. An Oslo subpopulation of heart transplant recipients (76 patients), assessed up to 36 months after transplant, presented decreasing viral loads after 6 months, with the lowest titers in the last observation [12].

## 7. Torque Teno Virus in Specific Transplanted Organs

TTV was assessed in numerous studies, predominantly with small numbers of investigated patients comprising different aspects of organ transplantation. The authors analyzed recipients of different organs or a mixed group of recipients (e.g., renal and liver recipients), at different time points after transplantation or observation, over the course of time. The most numerous and best investigated group were the NTx recipients. Authors of the studies used different molecular methods of viral load assessment. Even in the pretransplant period, some differences were visible; for example, in ventricular assist device patients, the observed viral load was higher [36]. Despite this fact, we tried to summarize the most important studies and their results in terms of immunosuppression optimization in different organ recipients. The details are presented in Table 1.

A detailed presentation of the role of TTV in particular organ transplants is presented below.

### 7.1. Torque Teno Virus in Renal Graft Recipients

Renal graft recipients, as they represent the most numerous group among graft recipients, are the most investigated group in terms of TTV analysis. It was shown that TTV replication is higher in cases of patients with no confirmed humoral or cellular rejection [37,38]. According to research on immunosuppressive drugs, higher levels of immunosuppression and more potent immunosuppressive drugs in most studies are connected with higher TTV loads in renal graft recipients [17]. Meta-analyses have consistently demonstrated a potential correlation between the TTV and the incidence of opportunistic infections [14,39]. Studies are underway to assess the usefulness of TTV-guided immunosuppression optimization in NTx recipients [40].

### 7.2. Torque Teno Virus in Other Organ Transplant Graft Recipients

Lung and liver graft recipients, alongside NTx patients, constitute the most extensively studied groups among solid organ recipients concerning TTV infection. Several studies have included mixed cohorts, such as kidney and lungs [41] or kidney and liver [10]. These investigations have demonstrated that TTV exhibits a similar behavioral pattern to that observed in NTx recipients. Heart transplant recipients are not as extensively investigated as other SOT recipients, but available results seem to be promising [36].

**Table 1 viruses-16-00017-t001:** Torque Teno Virus studies on immunosuppression optimization in different organ recipients.

Author, Year	Population, No of Patients	TTV Load 1 Month after Tx	3 Months after Tx	Mean Time after Tx	TTV Measurement, Mean	Influence on Opportunistic Infections	ACR	ABMR	TTV Correlation with Immunosuppressive Drugs	Correlation with C0 Immunosuppressive Drug Levels	Factors Correlating with TTV	Method of Viral Assessment
Martin Schiemann M et al., 2017 [37]	715 kidney graft recipients			6.3 years	5.36 log_10_ copies/mL (IQR: 4.28–5.28)	NA	NA	Negative correlation: DSA MFI, C4d	TTV load, increased: tacrolimus, mTOR inhibitor, belatacept, triple immunosuppression, steroids decreased: cyclosporine	NA	TTV load higher: older age, male gender, HLA mismatch, no ABO incompatibility higher GFR	NucliSENS easyMAG platform (bioMerieux, Craponne, France)
Görtzer I, et al., 2015 [11]	46 lung transplant recipients			9.4, IQR: 7.6–10.7 log_10_ copies/mL, median 67 (41–92) days		NA			Tacrolimus—No quantitative correlation confirmed			NucliSENS easyMAG platform (bioMerieux, France)
Cañamero L et al., 2023 [17]	54 kidney transplant recipients	2.81 log_10_ copies/mL	6.73 log_10_ copies/mL			NA	Non-significant		No quantitative correlation confirmed, Induction use—positive correlation		Male gender (baseline), opportunistic infections	TaqMan (TM)-PCR assay human TTV APP2XDMP (Thermo Fisher, Life Technologies, Paisley, UK) in a StepOnePlus real-time PCR system (AB Applied Biosystems, Singapore).
Maggi F et al., 2018 [10]	280 kidney and liver recipients	5.0 ± 0.2(kidney), 4.7 ± 0.1(liver) log_10_ copies/mL	6.9 ± 0.2(kidney), 6.1 ± 0.2 (liver) log_10_ copies/mL		3.9 and (kidney) 4.2 (liver) log_10_ copies/mL	CMV reactivation- correlated			rATG- NS higher than basiliximab, cyclosporine higher than tacrolimus			TaqMan^®^-PCR *
Roberto P et al., 2023 [41]	146 kidney and 26 lung recipients		Lung transplant recipients: 5.9 log_10_ copies/mL, kidney: 4.8 log_10_ copies/mL				No correlation	NA			SARS-CoV-2 anti-Spike IgG antibodies formation higher in pts with lower TTV	CFX96 platform (Bio-Rad Laboratories, Inc., Hercules, CA, USA), real-time PCR
Simonetta F et al., 2017 [42]	39 liver transplant recipients				6 months: 6.3, IQR:1.4–8.73 log_10_ copies/mL 12 months: 5.34, IQR: 1.4–7.23 log_10_ copies/mL		Inverse correlation		NA	NA	Rejection activity index (inverse correlation)	RT-PCR *
Uyanik-Uenal K et al., 2020 [36]	50 heart transplant recipients		9.39 log_10_ copies/mL median 120 days (92–185)						Tacrolimus positively correlated		Ventricular assist device— higher baseline TTV load	PCR *

* In-house test (real-time PCR) applied or no exact data on methodology available. Caution: for the studies performed with application of in-house tests, the TTV loads cannot be exactly compared and significant differences could exist between individual studies.

## 8. TTV and Immunosuppressive Drugs

It is supposed that immunocompromised patients will have higher loads of viral replication. The need for intensity of immunosuppression depends on the organ transplanted, and as the accepted rule, the immunosuppression is the strongest during the first year after transplantation [24]. Due to immunogenic potential of the graft and possible consequences of graft lost for the patient, the highest immunosuppression should be in lung, heart, and bowel transplant recipients. There are known beneficial effects of liver transplantation in the aspect of drug minimization [43]. There are even reports on long-term immunosuppression cessation in liver graft recipients [44]. In this context, the viral load is anticipated to be highest during the first year, particularly in patients receiving the most potent immunosuppression, encompassing both the types of applied immunosuppressive drugs and the targeted therapeutic drug levels. A reservoir of TTV is provided by the white blood cells. It was shown that T-cell function is inversely correlated with TTV load. In a study comprising allogenic hematopoietic stem cell recipients, an inverse correlation was observed between CD3^+^ T-cell proliferation capacity and TTV load (r = −0.39, *p* = 0.01) [45]. The CD3^+^ is located on T-lymphocytes, and is required for the T-cell activation process. The CD3^+^ T-receptor plays a pivotal role in the cellular rejection of solid organ transplants, and is the target of immunosuppressant therapies.

A strong and lymphodepleting drug used in organ transplantation is r-ATG. Similarly, mycophenolate mofetil is a substance that downregulates the proliferation of activated lymphocytes. Based on this fact, it was supposed that kinetics of TTV may be influenced by these two drugs. In a study on 280 liver and kidney recipients during the first year after transplantation, viremia fluctuated irrespective of the type of graft analyzed, but was connected with the type of immunosuppression used. Slightly higher viremia was associated with r-ATG induction when compared to basiliximab induction. In the cited study, higher viremia was also associated with cyclosporine regimen when compared to tacrolimus. The observation seems surprising considering the fact that the cyclosporine regimen is considered a weaker form of immunosuppression than tacrolimus. They also did not find the correlation between tacrolimus levels and virus load [10]. In turn, even short-term (5 weeks) mycophenolate mofetil cessation in 43 NTx patients caused a significant decrease in number of TTV copies. The intervention was part of a COVID-19 immunization protocol, and the authors also observed an inverse correlation between SARS-CoV-2 antibodies and TTV load [46].

In other studies performed on lung transplant recipients, virus load was higher in the case of tacrolimus regimen when compared to cyclosporine [29,47]. In 98 lung transplant patients assessed over 2 years after transplantation, the TTV loads were significantly higher in the group with tacrolimus from month 6 onward vs. cyclosporine (*p* < 0.001). Among the 98 lung transplant patients assessed over a 2-year period post transplantation, TTV loads exhibited a significant elevation in the tacrolimus-treated group compared to those receiving cyclosporine from the sixth month onward (*p* < 0.001) [29,47]. Tacrolimus instead of cyclosporine and the administration of triple-drug therapy were also associated with higher viral load in 715 NTx recipients [37]. Similarly, in living donor renal graft recipients, tacrolimus was associated with higher viral load than cyclosporine [9].

In 50 heart transplant recipients, the TTV load was correlated with the kind of calcineurin inhibitor used: as in the majority of available studies, tacrolimus application vs. cyclosporine caused higher viral loads [36]. However, contrary to the previous study, a two-center retrospective analysis involving 106 post-HTx patients did not identify a correlation between TTV and serum levels of immunosuppressants or episodes of rejection [12]. Underway is a multicenter, double-blinded, phase II trial assessing the role of TTV in guiding immunosuppression: the TTV-guideIT trial. The investigators compare noninferiority of TTV-guided immunosuppression in NTx recipients [40].

## 9. Role in Graft Rejection

A high level of immunosuppression is correlated with intense TTV replication. Typically, higher levels of immunosuppression are correlated with lower rejection risk [39]. It seems that a higher number of copies should be correlated with lower humoral and cellular rejection. There are some studies analyzing the correlation between the number of TTV copies and rejection risk, both humoral and cellular. The number of TTV copies was negatively correlated with the occurrence of antibody-mediated rejection in renal graft recipients. In a large analysis comprising 715 patients, and with a mean of 6.3 years after transplant, DSAs were assessed. Patients with DSAs present (86 pts.) underwent graft biopsy. It was stated that in patients with confirmed antibody-mediated rejection in the biopsy, the TTV level was only 25% of the level in patients without antibody-mediated rejection. This observation remained significant in a generalized linear model after adjusting for potential confounders (RR: 0.94; CI 95%: 0.0–099). Furthermore, the TTV level corelated inversely with C4d staining in biopsies [37]. In a prospective cohort study performed by Querido et al. comprising 81 NTx patients, the difference in TTV loads was also seen in terms of de novo DSAs. Patients at 12 months after transplantation with de novo antibodies had significantly lower TTV loads vs. patients who did not develop antibodies (3.7 vs. 5.3 log_10_ copies/mL, *p* < 0.05) [16]. The risk of graft rejection during the first 2 years after transplantation in a small (66 pts) group of NTx patients was correlated with a viral load at 1 month lower than 4.2 log_10_ copies/mL [38]. In a study comprising renal graft recipients, a viral load < 1 log_10_ was associated with the risk of biopsy-proven acute cellular graft rejection [48]. In a cohort of 389 recipients of living NTx donors, a predictive model for acute rejection risk based on TTV load was established. The study revealed a notable inverse association between time to rejection and TTV load: for each 10-fold increase (1 log) in TTV load, the risk of rejection decreased by 0.74 (*p* < 0.001) [9]. A meta-analysis comprising NTx recipients showed a significantly lower viral load in patients with biopsy-proven episodes of acute rejection [14]. A similar observation was confirmed in 39 liver transplant recipients: a lower occurrence of acute cellular rejection was reported in patients with higher TTV loads [42].

Among 34 lung transplant recipients evaluated within the initial year post transplantation, a significant reduction in TTV levels was observed preceding biopsy-proven rejection. The sensitivity for a 10-fold decrease in TTV load in predicting rejection was 0.74, with a specificity of 0.99 [49]. In a study involving 106 heart transplant recipients categorized based on the occurrence of any episode of acute cellular rejection (grade ≥ 2) within the initial 12 months post transplantation, no significant differences were detected concerning TTV load across assessed intervals. It is noteworthily that the authors did not evaluate or compare the TTV load prior to confirmed episodes of rejection [12]. Considering the variability in TTV load dynamics under diverse clinical conditions, such as during ongoing rejection episodes, disparities might exist when employing alternative assessment methods.

## 10. Torque Teno Virus and Recipient Age

In a population of 313 healthy, immunocompetent people, it was shown that TTV load is connected with age. Young people have lower TTV loads than middle-aged individuals, and the middle-aged lower than the elderly [50]. This seems connected with normal ageing process of immune system. Studies on transplant recipients showed a higher viral load in older patients [37]. A single-center study comprising 169 consecutive NTx recipients elucidated higher TTV loads in older (>56 years old) recipients and older donors [50]. However, the results might be influenced by additional factors that are correlated with age. In contrast, in a study comprising lung recipients, the correlation between age and viral load was not confirmed [11].

## 11. Torque Teno Virus Role in Carcinogenesis of Graft Recipients

According to the current meta-analysis [51] comprising 34 articles and 2145 investigated patients, the role of TTV in carcinogenesis is still to be defined. Considering solid organ recipients, a recent study on bladder cancer comprising 43 solid organ recipients reported that the viruses under investigation (JC polyomavirus, BKPyV, papillomaviruses, TTV) were present in almost 50% of cancer biopsies, and among them, the TTV was present in 12% of biopsies. The authors claim that the role and causality in carcinogenesis is not known [52].

## 12. Conclusions

TTV seems to be a novel promising marker in assessing immunocompetence status in solid organ recipients. As an additional parameter, it might have utility in noninvasive rejection and infection risk monitoring. New technologies such as gene expression profiling and dd-cfDNA, in combination with immunosuppressive drug monitoring, seem to benefit from TTV load assessment.

## Figures and Tables

**Figure 1 viruses-16-00017-f001:**
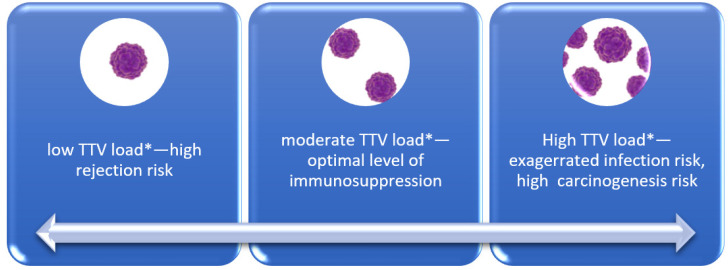
TTV level according to complications. * The optimal TTV load is still to be determined according to organ transplanted.

## Data Availability

Not applicable.

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
