# Peer review of "Torque Teno Virus (TTV)—A Potential Marker of Immunocompetence in Solid Organ Recipients"

_viruses, 2023, doi:10.3390/v16010017_

Round 1

Reviewer 1 Report

Comments and Suggestions for Authors

The review written by Kuczaj and collegues deals with „TTV - a Potential Marker of Immunocompetention in Solid Organ Recipients.“

 In this review the authors discuss studies on SOT patients in which plasma TTV load quantitation as a potential marker of immunosuppression has been investigated. In particular, the authors included studies that demonstrated the TTV load behaviour after SOT and the predictive value of the TTV load to determine the risk of experiencing a graft rejection (low TTV load - high risk), or opportunistic infections (high TTV load - high risk), or how effective the humoral immune response will be after vaccination (high TTV load - low response). Finally, the authors included two short sections dealing with the TTV load depending on the age of the graft recipients and whether TTV may also play a role in carcinogenesis.

This review gives a comprehensive overview of a number of recent studies with a strong focus on NTX patients. It is well-structured and it includes many findings and specific information. Also the table gives a detailed overview although in the current form it was not easy to read.

Here are my comments:

Title:

Line 2: do the authors mean „immunocompetence“ instead of immunocompetention?

1.Introduction:

I would suggest to revise this section in terms of the cited and missing references, conciseness, and correctness. For me, the studies mentioned are not always the most representative ones.

e.g.

Line 23: anelloviridae; include the recent reference for the adapted nomenclature (https://ictv.global/taxonomy; https://ictv.global/ictv/proposals/2007.075a-xxV.v4.Anelloviridae.pdf)

Line 24: non-enveloped

Line 25: reference is missing

Line 29: please include a more relevant reference; It can frequently be found in serum .... but also in many tisues...

Line 31,35: please include a more relevant reference

Line 34-37: mainly KTX studies are mentioned; I would suggest to include LuTx and HTX studies as well;

Line 51-55: references are missing

3.Available Methods...

Figure 1: I would suggest to change the picture of the virus; for me, this seems to be a corona virus, but TTV is a non-enveloped virus;

There are also other ongoing prospective studieswhich could be mentioned; e.g. Gottlieb et al., Viral load-guided immunosuppression after lung transplantation (VIGILung)-study protocol for a randomized controlled trial. Trials. 2021 Jan 11;22(1):48. doi: 10.1186/s13063-020-04985-w.

Table 1:

I would suggest to revise the column with the Method of viral assessment; In my opinion it is important to know whether it is an in-house test (mainly real-time PCR) or a comemrcially available test. For the studies performed with in-house tests the TTV loads can not be exactly compared and may substantially differ between the different studies. This should be included in this column and also in the text.

Lines 233-237: it depends not only on the kind of the immunosuppressive drug but also on the given dose; this could be included for clarification;

Author Response

Dear Editorial Team! Dear Reviewer!

Thank you for giving our team the opportunity to submit a revised draft of my manuscript. I am grateful to the reviewers for their insightful comments on the paper titled “Title Torque Teno Virus (TTV) - a Potential Marker of Immunocompetence in Solid Organ Recipients”.

We incorporated changes to reflect the suggestions provided. I have highlighted red the changes within the manuscript. I hope that the article will be more valuable and compact due to the introduced changes.Here is a point-by-point response to the reviewers’ comments and concerns.

Comments from Reviewer

Title:

Line 2: do the authors mean „immunocompetence“ instead of immunocompetention? -> corrected

1.Introduction:

I would suggest to revise this section in terms of the cited and missing references, conciseness, and correctness. For me, the studies mentioned are not always the most representative ones. -> corrected

e.g.

Line 23: anelloviridae; include the recent reference for the adapted nomenclature (https://ictv.global/taxonomy; https://ictv.global/ictv/proposals/2007.075a-xxV.v4.Anelloviridae.pdf) -> corrected

Line 24: non-enveloped-> corrected

Line 25: reference is missing-> corrected

Line 29: please include a more relevant reference; It can frequently be found in serum .... but also in many tisues... -> corrected

Line 31,35: please include a more relevant reference-> corrected

Line 34-37: mainly KTX studies are mentioned; I would suggest to include LuTx and HTX studies as well; -> corrected

Line 51-55: references are missing-> corrected

3.Available Methods... -> corrected

Figure 1: I would suggest to change the picture of the virus; for me, this seems to be a corona virus, but TTV is a non-enveloped virus; -> corrected

There are also other ongoing prospective studieswhich could be mentioned; e.g. Gottlieb et al., Viral load-guided immunosuppression after lung transplantation (VIGILung)-study protocol for a randomized controlled trial. Trials. 2021 Jan 11;22(1):48. doi: 10.1186/s13063-020-04985-w. -> corrected

Table 1:

I would suggest to revise the column with the Method of viral assessment; In my opinion it is important to know whether it is an in-house test (mainly real-time PCR) or a comemrcially available test. For the studies performed with in-house tests the TTV loads can not be exactly compared and may substantially differ between the different studies. This should be included in this column and also in the text. -> corrected

Lines 233-237: it depends not only on the kind of the immunosuppressive drug but also on the given dose; this could be included for clarification; -> corrected

I look forward to respond to any further questions and comments you may have.

Sincerely,

Authors

Reviewer 2 Report

Comments and Suggestions for Authors

Dear Editor

Thank you for the opportunity to comment on this review paper entitled “Torque Teno Virus (TTV) – a Potential Marker of Immunocompetention in Solid Organ Recipients”” by Agnieszka Kuczaj et al.

There are currently no available biomarkers which can guide the degree of immunosuppression in the individual SOT patient and therefore a combination of through levels, surveillance biopsies and the clinical course is used to guide the amount of immunosuppression. Finding suitable noninvasive biomarkers to navigate the amount of immunosuppression is therefore urgently needed.

The paper is a review of the current literature of TTV in SOT organized by organ, rejection/infection/cancer. The natural history of TTV and TTV related to specific immunosuppressive medications.

All in all, a thorough review which gives a nice overview of the current knowledge of TTV in SOT.

Some few specific comments:

1)      When comparing values of TTV with different studies it would be preferable if the units were the same or recalculated so it is easier to compare.

2)      Page 2-line 49. True we use drug levels to monitor, but we also use, clinical course (number and type of infections, rejections) and surveillance biopsies. Please elaborate

3)      2.Materials and methodes. I think the authors should be more specific in describing their search methodes and the numbere of papers found and rejected for various reasons. See Prisma: https://systematicreviewsjournal.biomedcentral.com/articles/10.1186/s13643-020-01542-z

4)      Page 5. Line 177 sih ?

This review is thorough and provides good and well organized insight to current knowledge of TTV and the SOT recipient. I find it worth considering for publication, but will need some minor revision as suggested

Comments on the Quality of English Language

Good

Author Response

Dear Editorial Team! Dear Reviewer!

Thank you for giving our team the opportunity to submit a revised draft of my manuscript. I am grateful to the reviewers for their insightful comments on the paper titled “Title Torque Teno Virus (TTV) - a Potential Marker of Immunocompetence in Solid Organ Recipients.”

We incorporated changes to reflect the suggestions provided. I have highlighted red the changes within the manuscript. I hope that the article will be more valuable and compact due to the introduced changes.

Here is a point-by-point response to the reviewer’s comments and concerns.

Comments from Reviewer

1) When comparing values of TTV with different studies it would be preferable if the units were the same or recalculated so it is easier to compare. -> recalculated

2)      Page 2-line 49. True we use drug levels to monitor, but we also use, clinical course (number and type of infections, rejections) and surveillance biopsies. Please elaborate - > corrected

3)Materials and methodes. I think the authors should be more specific in describing their search methodes and the numbere of papers found and rejected for various reasons. See Prisma: https://systematicreviewsjournal.biomedcentral.com/articles/10.1186/s13643-020-01542-z - > corrected

4) Page 5. Line 177 sih ? - > six corrected

I look forward to respond to any further questions and comments you may have.

Sincerely,

Authors
